# Co-infections of *Schistosoma mansoni* and *Helicobacter pylori* in school-aged populations and implication for management and control practices in Niger State, Nigeria

Muinah Fowora[1], Babatunde Adewale[2], Hammed Mogaji[2,3,4*], Ibilola Omolopo[1], Balogun Daniel[2], Abdulrahman Aladejana[5], De'Broski R. Herbert[6]

**1** Molecular Biology and Biotechnology Department, Nigerian Institute of Medical Research, Yaba, Lagos State, Nigeria, **2** Public Health and Epidemiology Department, Nigerian Institute of Medical Research, Yaba, Lagos, Nigeria, **3** Public Health Program, Department of Applied and Behavioral Sciences, College of Arts and Sciences, Marian University, Indianapolis, Indiana, United States of America, **4** Department of Research and Development, Mission To Save The Helpless, Ikeja, Lagos State, Nigeria, **5** Department of Animal and Environmental Biology, Federal University Oye-Ekiti, Oye-Ekiti, Ekiti State, Nigeria, **6** Department of Microbiology and Immunology, Tulane University School of Medicine, New Orleans, Louisiana, United States of America

\* mogajihammed@gmail.com

## Abstract

*Helicobacter pylori* and *Schistosoma mansoni* are two important pathogens that infect the gastrointestinal tract, could be acquired earlier in childhood, and can remain asymptomatic for a long period. While *S. mansoni* is been controlled within routine primary healthcare program, there are no control guidelines for *H. pylori.* Here, we investigate the co-infection pattern of both pathogens to support improvement of management strategies. Fresh stool samples were obtained from 299 school-aged children between ages 6–17 years and analyzed using real-time and conventional polymerase chain reaction (PCR) for the direct detection of *S. mansoni* and *H. pylori* respectively. An overall prevalence of 19.7% (4.5% − 24.1%) was recorded for *H. pylori*, and 34.4% (0% − 42.1%) for *S. mansoni,* while co-infection of both was 7.0% (0% − 8.1%). Infections were significantly different across the study communities for *S. mansoni* (p = 0.00) when compared to *H. pylori* (p = 0.38). There were, however, no significant association between infection and gender (p > 0.05), with the following odds of infection for *S. mansoni* (OR=0.69, 95% CI: 0.42, 1.12), *H. pylori* (OR=1.33, 95% CI: 0.75, 2.36), and combination of both infections (OR=1.31, 95% CI: 0.83, 2.09). By age category, children below 14 years were twice likely to be exposed to the combination of both infections; age 12–14 years (OR=1.9, 95% CI: 1.06, 3.44), and age 9–11 years (OR=1.96, 95% CI: 1.11, 3.48). But they were also less likely to be exposed to *S. mansoni*; age 12–14 years (OR=0.52, 95% CI: 0.27, 0.97), and age 9–11 years (OR=0.43, 95% CI: 0.23, 0.79). Our study highlights an unexpectedly high prevalence of *S. mansoni*, moderate prevalence of H. *pylori*, and very low coinfection

**Data availability statement:** The datasets used and/or analyzed are available as Supporting Information files.

**Funding:** This work received partial support from the Nigerian Institute of Medical Research Grant for studies on Neglected Tropical Diseases [Grant number 0000134].

**Competing interests:** The authors have declared that no competing interests exist.

prevalence for both. Our findings on *S. mansoni* are particularly concerning, suggesting potential challenges in control programs. It is therefore important to consider integrated interventions, including chemotherapy and improved water, sanitation, and hygiene (WASH) services.

## Introduction

Schistosomiasis is one of the most prevalent neglected tropical diseases in the world with more than 200 million persons affected worldwide across 78 countries, mostly in areas where access to water, sanitation and hygiene facilities are lacking [1–4]. The disease is caused by water-borne trematode parasites of the genus *Schistosoma*, with two major species of *Schistosoma*; the first is the *S. haematobium* which inhabits the vesicular and pelvic venous plexus of the bladder and causes urogenital schistosomiasis and the second is *S. mansoni* which is more often in the inferior mesenteric veins draining the large intestine and causes intestinal schistosomiasis [2–4]. Schistosomiasis is a focal disease, with urogenital schistosomiasis been more prevalent, and perhaps more studied in endemic settings compared to, intestinal schistosomiasis [5]. Intestinal schistosomiasis has severe acute complications associated with the intestine – abdominal pain, diarrhea, blood in the stool, and in more severe cases, could lead to chronic enlargement of the liver and spleen, a condition known as hepatosplenomegaly [5–8]. The pathologies are worsened among children because of the developing immunity, with already established evidence on anemia, stunting, protein-energy malnutrition, school absenteeism and reduced cognition [9–10].

People infected with *S. mansoni* may be infected with *Helicobacter pylori* (*H. pylori*), a Gram-negative bacterium that also colonizes the gastrointestinal tract [11,12]. *H. pylori* is the most common human bacterium, and causative agent of gastritis, peptic ulcer disease, mucosa associated lymphoid tissue lymphoma and gastric cancer [13]. This bacterium infects 50% of the world's population, and in Africa its prevalence reaches as high as 80% as the infection is acquired during childhood. [13]. The risk factors for *H. pylori* acquisition have been reported to be mainly due to overcrowding, to have infected siblings or parents and to unsafe water sources [13]. Both *H. pylori* and *S. mansoni* share similar risk as transmission is intertwined with poor socio-ecological and economic factors, primarily lack or sub-optimal access to water and sanitation facilities, and poor hygiene practices [7,13]. However, *H. pylori* are chiefly transmitted via the fecal-oral or oral-oral route often in childhood, unlike *S. mansoni* which requires cercariae skin penetration [7,8,13]. Nevertheless, both pathogens infect the gastrointestinal tract, could be acquired earlier in childhood, can remain asymptomatic for a long period [7,8,13]. In regions where co-infections occur, there are overlapping clinical manifestations, with extra-gastroduodenal manifestations such as chronic liver diseases and iron-deficiency anemia [14–15]. There are also speculations of altered disease pathology, either as synergistic or antagonistic interactions, which may influence treatment and clinical outcomes, and susceptibility to other diseases [16].

Efforts targeted at controlling schistosomiasis have been ongoing for over a decade in Africa, and other endemic settings with routine diagnosis and administration of large scale praziquantel medications [1–3], there are currently no guidelines targeted at diagnosing or controlling *H. pylori* infections within routine primary health care programs [13]. In Nigeria, the most populous country in Africa, several studies have reported the co-infection of *H. pylori* with intestinal parasites and soil transmitted helminths and its associated burden [17,18], but studies on co-infections of *H. pylori* and *S. mansoni* are scarce. This evidence is necessary to support improvement of management strategies. This study therefore reports the prevalence of *H. pylori* and *S. mansoni* co-infection among school-aged population, and explore associations based on age and sex characteristics across five study communities in Niger State of Nigeria.

## Materials and methods

### Study area, design and selection of sampling sites

This study was conducted between 1st and 30th of April 2019, as part of a larger project, with the area and selection procedure previously described [19]. Briefly, five communities (Monai, Tamanai, Koro, Musarwa, and Yuna) perceived to be at risk of water-borne infections were purposively selected based on proximity to Kainji Dam in Borgu LGA of Niger state. Majority of the populations in these communities engage in farming (36.3%), fishing (12.9%), and trading (7.1%) [19]. In terms of access to access to Water, Sanitation and Hygiene (WASH), only 31% have access to improved sources of water and latrine facilities, and about 47% walk barefooted [19]. We employed a cross-sectional approach to administer pretested questionnaires and obtain fresh stool sample collection from school-aged children between ages 6–16 years.

### Sample size determination and selection of study participants

Participants recruited in this study were subset of those recruited in the larger study [19]. Four hundred and ten pupils living in communities close to Kainji Dam were enlisted following their informed consent in the larger project. The sample size was calculated followed WHO guidelines for recruiting a minimum of 50 pupils (above age 5) per school during helminthiasis survey [11]. The sample size was adjusted by 60% in each of the communities to cater for non-enrolled school-aged children. As such, a minimum of 400 children was targeted across the five selected communities. However, the recruitment of participants was in variant with the estimated sample size for some communities because of number of eligible participants who presented themselves for the study. Children within the age range of 6–17 years, who had not received any anti-helminth treatment within the last 6 months, were recruited from the communities for the study. Of the 410 children enrolled in the larger study, only 22/133 from Monai, 79/79 from Tamanai, 124/124 from Koro, 36/36 from Musawa, and 38/38 from Yuna, consented to participate in this study. Information on the age and sex of children was collected using simple field forms.

### Questionnaire administration

Each pupil was allotted a unique code of six digits representing community, class and serial numbers. Demographic data such as name, age, sex were recorded using a simple field. The assigned participants ID were inscribed on both consent/ assent form, field form and stool specimen bottle. The field form was designed in the English language but administered in Hausa language.

### Collection of stool samples

Participants were provided with one sterile specimen bottle pre-labelled with their unique identification number, an applicator stick, a plain sheet of paper, and tissue paper to clean their anus. Participants were instructed to defecate on the plain sheet of paper and use the applicator stick to transfer a fresh portion into the specimen bottle. Specimen bottles were retrieved within 1 h of distribution, but collection was between 10 h and 14 h. All collected stool samples were sorted and

transported in iceboxes for processing within 2 hours of collection to the Nigerian Institute of Medical Research (NIMR) outstation laboratory located in Kainji.

### Laboratory processing of stool samples

The endemicity was classified based on the aggregated prevalence in each community. For H. pylori identification, stored stool sample in ice-boxes were transported to the Molecular Biology and Biotechnology Department of the Nigerian Institute of Medical Research. Genomic DNA was extracted and purified from the stool samples using NIMR Biotech stool DNA extraction kit in accordance with the manufacturer's instructions. An initial Polymerase Chain Reaction (PCR) was carried out using *Helicobacter* species specific primers as previously described [20]. All samples positive for the presence of *Helicobacter* species were further analyzed with a second PCR targeting the 16S rRNA gene of *H pylori*. The second PCR reaction was carried out using the primer set Hpy-F (5′-CGC ACC TGC TGG AAC ATT AC-3′) and Hpy-R (5′-CGT TAG CTG CAT TAC TGG AGA-3′) as previously described [21]. All PCR fragments were separated on a 2% agarose gel and visualized after ethidium bromide staining. A 138-bp fragment confirmed the presence of *H. pylori* in the stool sample.

A real time-PCR was used to detect cell-free *S. mansoni* in the DNA samples. The reaction was carried out using the set of primers and probes that amplifies a 121 bp tandem repeat sequence of *S. mansoni* strain SM 1–7 (GenBank accession number: M61098) as previously described [22,23]. The primer and probe sequence were Sm FW 5′-CCG ACC AAC CGT TCT ATG A-3′; Sm RV 5′-CAC GC TCT CG C AAA TAA TCT AAA-3′; Sm probe 5′-[FAM] TCG TTG TAT CTC CGA AAC CAC TGG ACG [(BHQ1])-3′ all synthesized by Biomers GMBH, Germany.

### Data management and analysis

Data obtained from this study were imported and analyzed in R. software version 4.3.2 (S1 File). Descriptive statistics, including frequencies and percentages, were used to describe the prevalence and co-infection patterns, and tested association across communities using chi-square statistics. Significant associations were established when $p < 0.05$. Furthermore, we performed three univariate logistic regression model each between single (*H. pylori, S. mansoni*), co-infection (*H pylori* and *S. mansoni*) as response variables, and demographic factor (sex and age) as covariates Regression estimates were reported in the form of odds ratio (OR) and 95% confidence interval (CI). The significance level was established as $p < 0.05$.

### Ethical statement and considerations

The study received approval from the Nigerian Institute of Medical Research ethical review board (IRB/18/042). Informed consent was obtained from parents/guardian and children. Children's assent was obtained verbally and documented via thumb print through an assent form in the presence of the parent or legal guardian who provided written consent. Unique identifiers and a password-protected database were also used to ensure anonymity and confidentiality through the study procedures. The study procedures were implemented following the ethical standards of the Helsinki Declaration (1964, amended most recently in 2008) of the World Medical Association. STROBE checklist for cross-sectional studies (S2 File) and Inclusivity checklist (S3 File) have been attached as supplementary files.

## Results

### Demographic characteristics of study participants

Table 1 presents the age and sex distribution of the 299 participants recruited for the study from five communities: Monai (n = 22, 7.4%), Tamani (n = 79, 26.4%), Koro (n = 124, 41.5%), Musawa (n = 36, 12.0%), and Yuna (n = 38, 12.7%). Male participants (n = 174, 58.2%) were more than females (n = 125, 41.8%). The age group distribution showed that most participants were aged 9–11 years (n = 102, 34.1%), followed by 6–8 years (n = 96, 32.1%), 12–14 years (n = 90, 30.1%), and

**Table 1. Sex and age characteristics of the study population.**

| | | Communities | | | | | |
|---|---|---|---|---|---|---|---|
| | N = 299 | Monai (N = 22) | Tamani (N = 79) | Koro (N = 124) | Musawa (N = 36) | Yuna (N = 38) | p-value |
| **age** | | | | | | | 0.00 |
| 6-8yrs | 96 (32.1%) | 21 (95.5%) | 5 (6.3%) | 41 (33.1%) | 22 (61.1%) | 7 (18.4%) | |
| 9-11yrs | 102 (34.1%) | 1 (4.5%) | 43 (54.4%) | 38 (30.6%) | 14 (38.9%) | 6 (15.8%) | |
| 12-14yrs | 90 (30.1%) | 0 (0.0%) | 26 (32.9%) | 39 (31.5%) | 0 (0.0%) | 25 (65.8%) | |
| 15-17yrs | 11 (3.7%) | 0 (0.0%) | 5 (6.3%) | 6 (4.8%) | 0 (0.0%) | 0 (0.0%) | |
| **sex** | | | | | | | 0.00 |
| Female | 125 (41.8%) | 8 (36.4%) | 41 (51.9%) | 56 (45.2%) | 16 (44.4%) | 4 (10.5%) | |
| Male | 174 (58.2%) | 14 (63.6%) | 38 (48.1%) | 68 (54.8%) | 20 (55.6%) | 34 (89.5%) | |

N: Number of participants; Values correspond to N (%).

15–17 years (n = 11, 3.7%). Significant differences were observed in the age and sex distribution across the communities (p = 0.00). Monai presented the highest proportion of participants aged 6–8 years (21, 95.5%), while Yuna had the highest proportion in the 12–14 years category (25, 65.8%). Males were significantly more represented in Yuna (34, 89.5%) compared to other communities (p = 0.00).

Table 2 summarizes the prevalence of *H. pylori and S. mansoni* infections among the participants. An overall prevalence of 19.7% was recorded for *H. pylori* across the communities. Tamani (24.1%) had the highest prevalence, while Monai (4.5%) had the lowest prevalence. However, there were no statistically significant differences in *H. pylori* prevalence across the communities (p = 0.38). In contrast, *S. mansoni* had a significantly higher overall prevalence of 34.4%. Yuna (42.1%), closely followed by Koro (41.1%) had the highest prevalence across communities, while Monai reported no cases (0.0%). The differences in *S. mansoni* prevalence across communities were statistically significant (p = 0.00). Participants were further categorized based on their infection status: no infection (52.8%), mono-infection (40.1%), or co-infection (7.0%). Co-infection rates were highest in Koro (8.1%), followed by Yuna (7.9%), while Monai had (0%) coinfection rate. These results were statistically different among the communities (p = 0.00).

Table 3 shows the association between infections and demographic variables. Compared to females, males had non-significant associations with *H. pylori* (OR: 1.33, 95% CI: 0.75–2.36, p = 0.33), *S. mansoni* (OR: 0.69, 95% CI: 0.42–1.12, p = 0.14) and co-infection (OR: 1.31, 95% CI: 0.83–2.09, p = 0.25). By age, participants aged 9–11 and 12–14years had significant lower odds of infection with *S. mansoni* (OR: 0.43, 95% CI: 0.23–0.79, p = 0.01) and (OR: 0.52, 95% CI: 0.27–0.97, p = 0.04), respectively when compared to those aged 6–8 years. However, the odds of co-infection with *H. pylori and S. mansoni* were significantly higher among participants aged 9–11 years (OR: 1.96, 95% CI: 1.11–3.48, p = 0.02) and 12–14 years (OR: 1.90, 95% CI: 1.06–3.44, p = 0.03) compared to those aged 6–8 years.

## Discussion

Foremost, the overall prevalence recorded for *H. pylori* was 19.7%, which is similar to those reported in Owerri (20%) [24], but much lower than those reported in Imo State (72%) [25], Kano (70% and 38%) [26,27]. The northern part of Nigeria has been known to be heavily impacted [28,29], owing to increased risk associated with overcrowding and limited access to water and sanitation facilities [24,25]. This report, however, deviates from this notion, and disparities could be attributed to differences in diagnostic tools, sampling sites and study participants, or in fact, the hygiene standards of the latter two factors.

We observed *H. pylori* prevalence was particularly low in Monai compared to other communities which could suggest a difference in community-level sanitation practices [19], or health interventions, or maybe linked to

**Table 2. Prevalence of *H. pylori* and *S. mansoni* infection among the study population.**

|  | N = 299 | Monai (N = 22) | Tamani (N = 79) | Koro (N = 124) | Musawa (N = 36) | Yuna (N = 38) | p-value |
|---|---|---|---|---|---|---|---|
| ***H. pylori*** |  |  |  |  |  |  | 0.38 |
| Positive | 59 (19.7%) | 1 (4.5%) | 19 (24.1%) | 24 (19.4%) | 7 (19.4%) | 8 (21.1%) |  |
| Negative | 240 (80.3%) | 21 (95.5%) | 60 (75.9%) | 100 (80.6%) | 29 (80.6%) | 30 (78.9%) |  |
| ***S. mansoni*** |  |  |  |  |  |  | 0.00 |
| Positive | 103 (34.4%) | 0 (0.0%) | 29 (36.7%) | 51 (41.1%) | 7 (19.4%) | 16 (42.1%) |  |
| Negative | 196 (65.6%) | 22 (100.0%) | 50 (63.3%) | 73 (58.9%) | 29 (80.6%) | 22 (57.9%) |  |
| **Category** |  |  |  |  |  |  | 0.00 |
| None | 158 (52.8%) | 21 (95.5%) | 37 (46.8%) | 59 (47.6%) | 24 (66.7%) | 17 (44.7%) |  |
| Mono | 120 (40.1%) | 1 (4.5%) | 36 (45.6%) | 55 (44.4%) | 10 (27.8%) | 18 (47.4%) |  |
| Double | 21 (7.0%) | 0 (0.0%) | 6 (7.6%) | 10 (8.1%) | 2 (5.6%) | 3 (7.9%) |  |

N: Number of participants; Values correspond to N (%).

**Table 3. Association between infections and participants demographic.**

|  | Pathogens | | | | | |
|---|---|---|---|---|---|---|
|  | *H. pylori* | | *S. mansoni* | | Co-infection | |
|  | OR (95% CI) | p-value | OR (95% CI) | p-value | OR (95% CI) | p-value |
| **sex, N (%)** |  |  |  |  |  |  |
| Female | 1 |  | 1 |  | 1 |  |
| Male | 1.33 (0.75, 2.36) | 0.33 | 0.69 (0.42, 1.12) | 0.14 | 1.31 (0.83, 2.09) | 0.25 |
| **age, N(%)** |  |  |  |  |  |  |
| 6-8yrs | 1 |  | 1 |  | 1 |  |
| 9-11yrs | 0.77(0.37, 1.58) | 0.48 | 0.43(0.23, 0.79) | 0.01 | 1.96 (1.11, 3.48) | 0.02 |
| 12-14yrs | 0.75 (0.35, 1.56) | 0.44 | 0.52 (0.27, 0.97) | 0.04 | 1.9 (1.06, 3.44) | 0.03 |
| 15-17yrs | 0.53 (0.14, 2.63) | 0.39 | 0.84 (0.22, 4.07) | 0.81 | 1.45 (0.39, 5.36) | 0.56 |

N: Number of participants; OR: Odds Ratio; CI: Confidence Interval.

age, as about 96% of those in Monai were below age 8years. However, we couldn't find any association between mono-infections with *H. pylori* based on age. We presume our limited sample-size and covariate frame might have impacted the identification of any meaningful association. However, our regression model (for combined infection) further emphasized increased chance of infection in children aged 12–14years compared to those between age 9–11years. Previous studies have reported in consonance that the chance of infections is increased among older participants [25,26,30,31]. A possible explanation for the higher prevalence of infection among children aged 12–14 years is that exposures and reinfection are likely to persist in areas where WASH services are limited and socio-economic status are low, both conditions which reflect the nature of our study location [19,30–32].

For *S. mansoni*, an overall prevalence of 34.4% (19.4% − 42.1%) was recorded, this is quite high considering previous research findings in Nigeria — 2.93% in Sokoto [33], 8.93% in Kano [34], 3.6% in Ogun [35], and 9% in Osun [36]. This presents a source of concern for control programs with questions on persistent transmission considering environmental suitability and ineffective control program [37,38], or on a more worrisome note, the efficacy of praziquantel on *S. mansoni* [39]. Infections also shared similar patterns with *H. pylori* mono infection and with literature [35,38,40], as children aged 12–14 years were more infected, and none of the children aged 6–8 years in Monai were infected. The odds of infection,

even though lower in younger children were seen to progressively increase in magnitude as children age. These findings are logical, as infection with schistosomes is related to behavioral patterns, most especially water contact practices which are more common among older children due to increased liberty to engage in risky behaviors, such as swimming, bathing and fetching in river bodies infested with *Biomphalaria* snail species [35,38,40].

Our observation of a 7.0% co-infection rate is less concerning. However, considering the efforts targeted controlling schistosomiasis over the last decade, the high prevalence of *S. mansoni* alone was surprising and raises significant public health concerns, underscoring the need for enhanced chemotherapy and improved water, sanitation, and hygiene (WASH) services. Reports of synergistic and antagonistic immunological interactions between *S. mansoni* and *H. pylori* [16,41] prompt further questions about whether these pathogens might exhibit antagonistic relationships at the ecological or behavioral levels, potentially influencing transmission dynamics. This could explain the lower-than-expected co-infection prevalence observed in this study. While our findings are not robust enough to conclusively address this hypothesis, the question merits further investigation to better understand the interplay between these two pathogens in co-endemic settings. In this study, we have reported the prevalence of co-infection with *H. pylori and S. mansoni* infections among school-aged children in five communities in Northern Nigeria. Our findings are useful in guiding more robust research and programmatic actions.

## Conclusion

Our study highlights an unexpectedly high prevalence of *S. mansoni*, moderate prevalence of H. *pylori*, and very low coinfection prevalence for both. Our findings on *S. mansoni* is particularly concerning, suggesting potential challenges in control programs, that needs investigation. While the co-infection rate was lower than expected, the findings still underscore the need for integrated interventions, including chemotherapy and improved water, sanitation, and hygiene (WASH) services, to address both infections effectively, especially among older children.

### Study limitations

A key limitation of this study is the restricted number of covariates included in the analysis. While the study focused on exploring associations by age and gender, a more comprehensive predictive model incorporating additional covariates could provide a deeper understanding of factors influencing co-infection patterns. Future studies should consider including a broader range of demographic, environmental, and behavioral variables to enhance model robustness and better account for potential confounding effects.

## Supporting information

**S1 File. Dataset.**
(XLSX)

**S2 File. STROBE checklist cross sectional.**
(DOCX)

**S3 File. Inclusivity in global research questionnaire.**
(DOCX)

## Acknowledgments

We are grateful to the community leaders across the study communities, and all the health workers who facilitated the implementation process.

## Author contributions

**Conceptualization:** Muinah Fowora, Babatunde Adewale, Hammed Mogaji, De'Broski R. Herbert.

**Formal analysis:** Muinah Fowora, Babatunde Adewale, Hammed Mogaji, Ibilola Omolopo.

**Funding acquisition:** Babatunde Adewale.

**Investigation:** Muinah Fowora, Babatunde Adewale, Hammed Mogaji, Ibilola Omolopo, Balogun Daniel, Abdulrahman Aladejana, De'Broski R. Herbert.

**Methodology:** Muinah Fowora, Babatunde Adewale, Hammed Mogaji, Ibilola Omolopo, Balogun Daniel, Abdulrahman Aladejana, De'Broski R. Herbert.

**Project administration:** Babatunde Adewale.

**Resources:** Muinah Fowora, Babatunde Adewale, Hammed Mogaji.

**Software:** Hammed Mogaji.

**Supervision:** Muinah Fowora, Babatunde Adewale, Hammed Mogaji.

**Validation:** Muinah Fowora, Babatunde Adewale, Hammed Mogaji, De'Broski R. Herbert.

**Visualization:** Hammed Mogaji.

**Writing – original draft:** Muinah Fowora, Babatunde Adewale, Hammed Mogaji, Balogun Daniel, Abdulrahman Aladejana, De'Broski R. Herbert.

**Writing – review & editing:** Muinah Fowora, Babatunde Adewale, Hammed Mogaji, Ibilola Omolopo, De'Broski R. Herbert.

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
