## [Decision Letter · Decision Letter 0]

26 Aug 2025

PONE-D-25-40863Co-infections of Schistosoma mansoni and Helicobacter pylori in School-Aged Populations and implication for management and control practicesPLOS ONE

Dear Dr. Mogaji,

Thank you for submitting your manuscript to PLOS ONE. After careful consideration, we feel that it has merit but does not fully meet PLOS ONE’s publication criteria as it currently stands. Therefore, we invite you to submit a revised version of the manuscript that addresses the points raised during the review process. 

We look forward to receiving your revised manuscript.

Kind regards,

Clement Ameh Yaro, Ph.D

Academic Editor

PLOS ONE

Journal Requirements:

This work received partial support from the Nigerian Institute of Medical Research Grant for studies on Neglected Tropical Diseases [Grant number 0000134].

4. Please note that funding information should not appear in any section or other areas of your manuscript. We will only publish funding information present in the Funding Statement section of the online submission form. Please remove any funding-related text from the manuscript.

5. Please amend your authorship list in your manuscript file to include author Hammed Oladeji Mogaji.

6. Please amend the manuscript submission data (via Edit Submission) to include author Hammed Mogaji.

7. Please remove all personal information, ensure that the data shared are in accordance with participant consent, and re-upload a fully anonymized data set.

Reviewers' comments:

Reviewer's Responses to Questions

**Comments to the Author**

1. Is the manuscript technically sound, and do the data support the conclusions?

Reviewer #1: Yes

Reviewer #2: Partly

2. Has the statistical analysis been performed appropriately and rigorously? 

Reviewer #1: Yes

Reviewer #2: No

3. Have the authors made all data underlying the findings in their manuscript fully available?

Reviewer #1: Yes

Reviewer #2: No

4. Is the manuscript presented in an intelligible fashion and written in standard English?

Reviewer #1: Yes

Reviewer #2: Yes

5. Review Comments to the Author

Reviewer #1: The manuscript looks good. The study of the coinfections of Schistosoma mansoni and Helicobacter pylori is a good one and worthy of investigation as the two organisms cohabit gastrointestinal track. This has provided useful information and better understanding on how they impact diseases etiology and this has great public health value and policy implications.

The authors should clarify and reconcile the data presented in the manuscript. They should address all comments made on the manuscript.

Reviewer #2: Study overview

This study attempts to evaluate co-infections of Schistosoma mansoni and Helicobacter pylori among school-aged populations across five communities. Although, I find this initiative timely and important, the manuscript does not satisfactorily pitch the link between the two pathogens in the introduction, despite reporting findings that could possibly inform future interventions.

I hope my somewhat extensive comments do not discourage the authors and that they are received in the constructive and supportive spirit in which they are offered.

For clarity, I have divided my comments into three categories, which I hope should help improve the overall presentation of your manuscript.

General comments:

1. It is not clear why these two pathogens are studied together. The manuscript does not precisely describe whether they share transmission pathway, immunological interaction etc. I expected to see a clearer rationale linking the topic and objective.

2. Also, I expected to read key findings that could inform changes in the management of both pathogens.

3. In the Methods section, the collection of data on water contact activities and parent’s occupation was mentioned. However, these variables were neither included in the analysis nor reported in the Results and Discussion. Their omission, along with the absence of other important variables such as proxies for overcrowding and WASH indicators, somewhat weakens the manuscript, as such factors are crucial for guiding interventions.

4. To emphasize the importance of this finding, consider noting in the Discussion (perhaps in the last paragraph) that schistosomiasis is classified as a Neglected Tropical Disease (NTD) according to the World Health Organization (WHO). This will highlight the importance of your work in identifying the locations and age groups with the highest prevalence odds of infection, thereby providing evidence to guide targeted interventions.

Major comments

1. The introduction reads a bit underdeveloped. The literature review should be expanded to situate the study within existing knowledge. For example, lines 74-76 should more robustly explain the “similar risk” described. While both pathogens are WASH-related GIT, H. pylori is chiefly fecal-oral, oral-oral route often in childhood, while S. mansoni is via skin penetration by cercariae in contaminated water – both possessing different risk factors. As such, significant improvement is need in the introduction to satisfactorily present the study especially given what the study intend to present.

2. Lines 234-236: Given that H. pylori infection is commonly reported during childhood and increases with age, the 1% seropositivity reported in Monai may be better explained by other unreported variables. Since these were omitted, the title should perhaps be revised to reflect the actual scope (demographic-related), rather than implying analyses that were not performed.

3. Consider starting your Discussion section with the presentation of your “principal findings”, as this would enhance the quality of your manuscript. This way, you could move the first two sentences under Discussion to the latter part of the manuscript.

4. Lines 240 and 254: The text suggests a higher risk of co-infection among older children, which is inconsistent with both the study design and findings. The measure used in this study is OR, which captures “odds”, and not “risk”. Also, "Older children" is vague and not previously defined. The statement also overlooks the fact that no statistically significant evidence was found for the 15-17 age group. I recommend replacing "risk" with "chance". If the term "younger children" and "older children" are to be retained, age ranges should be defined in the Methods section. Otherwise, refer to the age ranges in the Discussion as they appear in the tables.

Minor comments

1. Keywords: It would be advisable to select keywords that do not overlap with the words in the title of the article, to improve discoverability during search.

2. Lines 24- 26: The statement appears awkward and incomplete, lacking logical flow between the two pathogens. The authors should attempt to establish explicit mechanistic link explaining why these pathogens are being co-investigated.

3. Consider moving the ethical state to the final part of Methods, as is the common practice

4. The phrase “Niger area” is ambiguous, are you referring to Niger State, the River Niger basin, or another region? Please clarify.

5. Please justify the use of real-time PCR for S. mansoni and conventional PCR for H. pylori, especially since PCR detection of H. pylori is less common than antigen or urea breath testing.

6. The abstract overly descriptive. To enrich it, kindly incorporate a clear discussion of the key findings and potential explanations.

7. Lines 101-103: Please specify what type of consent obtained and the database repository used.

8. Lines 112 – 114: Please provide a reference to support this claim.

9. There is a slight inconsistency in reported age of participants. E.g. the Methods states 6-16 yrs, but the Results report 6-17 yrs. Please address this discrepancy.

10. The sample size justification is vague, lacking essential parameters such as expected prevalence and margin of error.

11. Line 140: Were participants instructed to provide stool samples at a specific time of the day, or was any sample acceptable? Given the collection period, how was sample quality and integrity ensured? Why was repeated sampling not considered, since egg shedding S. mansoni can be intermittent?

12. Line 142: Define “NIMR” at the first mention.

13. Kindly ensure that all scientific names are properly italicized following general taxonomy rules. There are sporadic occasions where this was not done in the manuscript (e.g. lines 147 and 148). Also, remove italics from “and” in “H. pylori and S. mansoni infections” at lines 193 and 205.

14. Kindly consider adding a minimal description of the variables included in the models is missing. What covariates were used? The supplementary materials do not include a data dictionary explaining each variable, and this information should still be included in the Methods since readers may not fully understand what is being tested.

15. Generally, reporting both p-values and confidence intervals together is somewhat redundant (e.g., lines 174 and 175, as well as throughout the Results section). If opting to continue with p-values, kindly standardize it throughout the manuscript, presenting all to three 3 significant digits.

16. Line 183: The phrasing suggests a decreasing sequence, but this is not the case.

17. Add footnotes to tables defining all terms and abbreviations.

18. Line 220: It would be more appropriate and academically formal to use “Odds Ratio”.

19. Lines 224-225: Briefly specify what the “some evidence that might be important [...]” are

20. Line 227: “similar those”, please revise.

21. Lines 230-232: The justification here is unconvincing. Niger State is also in Northern Nigeria, and you have not conducted or presented any socio-environmental analysis to substantiate this claim.

22. While the use of ORs in cross-sectional studies is common, this being a prevalence study, results may risk overestimation. In this case, Prevalence Ratios (PRs) might provide a more accurate measure of association. However, I do not discourage the use of ORs, but it would be worthwhile to mention this as a limitation.

23. Line 260: What exactly was “previously predicted”? No hypotheses were explicitly stated earlier.

24. Per journal policy, please deposit the script and supporting data in an open repository.

6. PLOS authors have the option to publish the peer review history of their article (what does this mean?). If published, this will include your full peer review and any attached files.

Reviewer #1: **Yes: **Obiageli Josephine Nebe

Reviewer #2: No

---

## [Author Response · Author response to Decision Letter 1]

25 Sep 2025

REVIEWER 1

The manuscript looks good. The study of the coinfections of Schistosoma mansoni and Helicobacter pylori is a good one and worthy of investigation as the two organisms cohabit gastrointestinal track. This has provided useful information and better understanding on how they impact diseases etiology and this has great public health value and policy implications. The authors should clarify and reconcile the data presented in the manuscript. They should address all comments made on the manuscript.

Response; We are grateful for the very constructive and positive comments of the reviewer. We have now revised the manuscript taking into considerations all the reviewers comments and opinions

Comment 1: Full title is preferable.

Response: The title has been revised

Comment 2: It is not only MDA of praziquantel tablets that are being used for the elimination programme, rather other cross cutting approaches including Integrated Vector Management, WASH, Behavioural Change Communication are all part of the interventions that will facilitate elimination process.

Response: The abstract has also been revised accordingly. The new introductory lines now reads “Helicobacter pylori and Schistosoma mansoni are two important pathogens that infect the gastrointestinal tract, could be acquired earlier in childhood, and can remain asymptomatic for a long period. While S. mansoni is been controlled within routine primary healthcare program, there are no control guidelines for H. pylori. Here, we investigate the co-infection pattern of both pathogens to support improvement of management strategies..”

Comment 4: Does this refer to Niger basin or Niger State setting? kindly clarify.

Response: Yes, we have now revised the text to read “Niger State”

Comment 5: The findings for S. mansoni is quite disturbing as several rounds of MDAs have been implemented in Niger State. This calls to question the quality of intervention implemented all these years. The authors should have investigated the MDA history, therapeutic and geographical reach

Response: Yes, we agree absolutely with the reviewer, and noted this in our discussion. This observation is worth further investigation, and public health action.

Comments: Line 127-128: Why variation in the sample size

Response: We thank the reviewer for this important observation. This study is a sub-study, that follows a larger study where 410 children were recruited. However, for this sub-study only 73% consented to participate in this study. i.e. 100% compliance across all the communities except Monai (16.5% compliance). We have revised the sample-size estimation sub-section to provide more clarity.

Comments: Line 180-181: Kindly reconcile this number 299 with the earlier number of participants 410 under the smaple size

Response: We thank the reviewer for this important observation. This study is a sub-study, that follows a larger study where 410 children were recruited. However, for this sub-study only 73% consented to participate in this study. i.e. 100% compliance across all the communities except Monai (16.5% compliance). We have revised the sample-size estimation sub-section to provide more clarity.

Comments: Table 1, See comment above

Response: We thank the reviewer for this important observation. We have provided the response to this comment in our previous response.

Comments; Line 247-249: Kindly reconcile this statistics with the FMOH data base on schistosomiasis prevalence 2013-2017mappimg data. It is available with the National Programme and WHO ESPEN.

 Response: We thank the reviewer for this important observation. We have revised the text to emphasize our comparison with research findings..

REVIEWER 2:

C1: Study overview

This study attempts to evaluate co-infections of Schistosoma mansoni and Helicobacter pylori among school-aged populations across five communities. Although, I find this initiative timely and important, the manuscript does not satisfactorily pitch the link between the two pathogens in the introduction, despite reporting findings that could possibly inform future interventions. I hope my somewhat extensive comments do not discourage the authors and that they are received in the constructive and supportive spirit in which they are offered. For clarity, I have divided my comments into three categories, which I hope should help improve the overall presentation of your manuscript.

RESPONSE: We thank the reviewer for the comments, which have significantly improved the quality of our manuscript. We have carefully incorporated all the review suggestions made and hope you find the manuscript worthy of publication.

General comments:

1. It is not clear why these two pathogens are studied together. The manuscript does not precisely describe whether they share transmission pathway, immunological interaction etc. I expected to see a clearer rationale linking the topic and objective.

Response: We thank the reviewer for this comment. In our manuscript between lines 74-80, we describe the transmission pathway, clinical manifestations and speculations of altered disease pathology. We cited 6 references within this space. Further between lines 82-91, we further linked these lines with the rationale of generating evidence that improves the management of both diseases.

2. Also, I expected to read key findings that could inform changes in the management of both pathogens.

Response: We thank the reviewers for this comment. Following our analysis, we reported an unexpectedly high prevalence of S. mansoni, moderate prevalence of H. pylori, and very low coinfection prevalence for both. Our findings on S. mansoni is particularly concerning, suggesting potential challenges in control programs that needs investigation. While the co-infection rate was lower than expected, the findings still underscore the need for integrated interventions, including chemotherapy and improved water, sanitation, and hygiene (WASH) services, to address both infections effectively, especially among older children. These excerpts are currently in the abstract (lines; 32-43; results; discussion; conclusion (lines 271-277)

Comments 3. In the Methods section, the collection of data on water contact activities and parent’s occupation was mentioned. However, these variables were neither included in the analysis nor reported in the Results and Discussion. Their omission, along with the absence of other important variables such as proxies for overcrowding and WASH indicators, somewhat weakens the manuscript, as such factors are crucial for guiding interventions.

Response: We appreciate the reviewer for this observation and clearly acknowledge this gap. We have revised these lines by deleting the variables that were not collected. Also, we added the context of the five study communities between lines 113-115 based on the previous study conducted amongst this population. The poor WASH access in the population still justifies the recommendations given, since our participants were drawn out (i.e., subsets) of that locality. Please see lines 124-126, and lines 133-137.

Comment 4. To emphasize the importance of this finding, consider noting in the Discussion (perhaps in the last paragraph) that schistosomiasis is classified as a Neglected Tropical Disease (NTD) according to the World Health Organization (WHO). This will highlight the importance of your work in identifying the locations and age groups with the highest prevalence odds of infection, thereby providing evidence to guide targeted interventions.

Response: We thank the reviewer for this comment. We have added a line to the last paragraph (please see line 264), and also introduced schistosomiasis as a NTD in the first line of our introduction.

Major comments

Comment 1. The introduction reads a bit underdeveloped. The literature review should be expanded to situate the study within existing knowledge. For example, lines 74-76 should more robustly explain the “similar risk” described. While both pathogens are WASH-related GIT, H. pylori is chiefly fecal-oral, oral-oral route often in childhood, while S. mansoni is via skin penetration by cercariae in contaminated water – both possessing different risk factors. As such, significant improvement is need in the introduction to satisfactorily present the study especially given what the study intend to present.

Response: We appreciate the reviewer for this amazing comment, which was super useful. We have revised that paragraph to illustrate both the commonality and differences in their risk factors. Please see Line 75-79

Comments 2. Lines 234-236: Given that H. pylori infection is commonly reported during childhood and increases with age, the 1% seropositivity reported in Monai may be better explained by other unreported variables. Since these were omitted, the title should perhaps be revised to reflect the actual scope (demographic-related), rather than implying analyses that were not performed.

Response: We thank the reviewer for this comment. Considering the revisions we made, We have now provided the overall occupation, access to latrine and walking barefoot for these 5 communities studies. Based on this access to sanitation in Monai was 22%, and this resonates with the lines we have on Monai.

Comments 3. Consider starting your Discussion section with the presentation of your “principal findings”, as this would enhance the quality of your manuscript. This way, you could move the first two sentences under Discussion to the latter part of the manuscript.

Response: We thank the reviewer for this comment. We have now moved the two sentences to the latter part of the manuscript

Comments 4. Lines 240 and 254: The text suggests a higher risk of co-infection among older children, which is inconsistent with both the study design and findings. The measure used in this study is OR, which captures “odds”, and not “risk”. Also, "Older children" is vague and not previously defined. The statement also overlooks the fact that no statistically significant evidence was found for the 15-17 age group. I recommend replacing "risk" with "chance". If the term "younger children" and "older children" are to be retained, age ranges should be defined in the Methods section. Otherwise, refer to the age ranges in the Discussion as they appear in the tables.

Response: We thank the reviewer for this comment. This is such an amazing observation. We have revised the text completely to replace risk with chance, and also to replace the younger or older children with the exact age categories as they appear in the tables. Please see Lines 240-266.

Minor comments

Comments 1. Keywords: It would be advisable to select keywords that do not overlap with the words in the title of the article, to improve discoverability during search.

Response: We thank the reviewer for these comments. This is well noted

Comments 2. Lines 24- 26: The statement appears awkward and incomplete, lacking logical flow between the two pathogens. The authors should attempt to establish explicit mechanistic link explaining why these pathogens are being co-investigated.

Response: We thank the reviewer for these comments. We have now revised the abstract to portray clarity.

Comments 3. Consider moving the ethical state to the final part of Methods, as is the common practice

Response: We thank the reviewer for these comments. We have now moved it.

Comments 4. The phrase “Niger area” is ambiguous, are you referring to Niger State, the River Niger basin, or another region? Please clarify.

Response: We thank the reviewer for these comments. We have now corrected this. We meant Niger State.

Comments: 5. Please justify the use of real-time PCR for S. mansoni and conventional PCR for H. pylori, especially since PCR detection of H. pylori is less common than antigen or urea breath testing.

Response: We thank the reviewer for these comments. We agree that using H. pylori detection is less common than antigen or urea breath testing. However, we already had primers and other reagents for the molecular detection in our laboratory. In the face of limited funding for this work, we resulted to using PCR detection that was readily available in our laboratory. With regards to the use of qPCR for S. mansoni detection, studies have shown that RT-PCR was the most sensitive diagnostic method and gave the highest positive detection of Schistosoma species in stool specimen (Guegan et al. 2019). Hence, the use of RT-PCR for the detection of Schistosoma manosnii in this study.

Reference

Guegan H, Fillaux J, Charpentier E, Robert-Gangneux F, Chauvin P, Guemas E, et al. (2019) Real-time PCR for diagnosis of imported schistosomiasis. PLoS Negl Trop Dis 13(9): e0007711. https://doi.org/10.1371/journal.pntd.0007711

Comments; 6. The abstract overly descriptive. To enrich it, kindly incorporate a clear discussion of the key findings and potential explanations.

Response: We thank the reviewer for these comments. We have now revised the abstract to include these key findings as requested

Comments 7. Lines 101-103: Please specify what type of consent obtained and the database repository used.

Response: We thank the reviewer for this comment. Informed consent was obtained from parents/guardian and children. Children’s assent was obtained verbally and documented via thumb print through an assent form in the presence of the parent or legal guardian who provided written consent. We used a password protected computer to create and assign unique ID

Comments 8. Lines 112 – 114: Please provide a reference to support this claim.

Response: We thank the reviewer for these comments. We have now provided the clear metrics and also he reference for this

Comments 9. There is a slight inconsistency in reported age of participants. E.g. the Methods states 6-16 yrs, but the Results report 6-17 yrs. Please address this discrepancy.

Response: We thank the reviewer for these comments. We have now revised the text to 6-17 years. key findings as requested

Comments: 10. The sample size justification is vague, lacking essential parameters such as expected prevalence and margin of error.

Response: We thank the reviewer for these comments. The sample size estimation followed WHO guidelines of recruiting 50 pupils per school/villages. And we adjusted this size by 60% to cater for non-enrolled school aged children. However, of the 410 children recruited for the larger project, only 73% consented to participate in this study. i.e. 100% compliance across all the communities except Monai (16.5% compliance).

Comments: 11. Line 140: Were participants instructed to provide stool samples at a specific time of the day, or was any sample acceptable? Given the collection period, how was sample quality and integrity ensured? Why was repeated sampling not considered, since egg shedding S. mansoni can be intermittent?

Response; Thanks for the comments. Yes, only fresh samples were retrieved, as detailed between lines 164-166. Samples were collected within a very limited time frame of 1hr, and the parents/legal guardians supervised the voiding process. The need for sample integrity was sufficiently explained to the parents and their children.

Comments 12. Line 142: Define “NIMR” at the first mention.

Response; Thanks for the comments. This has now been addressed

13. Kindly ensure that all scientific names are properly italicized following general taxonomy rules. There are sporadic occasions where this was not done in the manuscript (e.g. lines 147 and 148). Also, remove italics from “and” in “H. pylori and S. mansoni infections” at lines 193 and 205.

Response; Thanks for the comments. This has now been addressed

Comments 14. Kindly consider adding a minimal description of the variables included in the models is missing. What covariates were used? The supplementary materials do not include a data dictionary explaining each variable, and this information should still be included in the Methods since readers may not fully understand what is being tested.

Response; Thanks for the comments. This has now been addressed

Comments 15. Generally, reporting both p-values and confidence intervals together is somewhat redundant (e.g., lines 174 and 175, as well as throughout

---

## [Decision Letter · Decision Letter 1]

19 Oct 2025

PONE-D-25-40863R1Co-infections of Schistosoma mansoni and Helicobacter pylori in School-Aged Populations and implication for management and control practices in Niger State, NigeriaPLOS ONE

Dear Dr. Mogaji,

Thank you for submitting your manuscript to PLOS ONE. After careful consideration, we feel that it has merit but does not fully meet PLOS ONE’s publication criteria as it currently stands. Therefore, we invite you to submit a revised version of the manuscript that addresses the points raised during the review process.

We look forward to receiving your revised manuscript.

Kind regards,

Clement Ameh Yaro, Ph.D

Academic Editor

PLOS ONE

**Journal Requirements:**

Reviewers' comments:

Reviewer's Responses to Questions

**Comments to the Author**

1. If the authors have adequately addressed your comments raised in a previous round of review and you feel that this manuscript is now acceptable for publication, you may indicate that here to bypass the “Comments to the Author” section, enter your conflict of interest statement in the “Confidential to Editor” section, and submit your "Accept" recommendation.

Reviewer #1: All comments have been addressed

Reviewer #2: (No Response)

2. Is the manuscript technically sound, and do the data support the conclusions?

Reviewer #1: Yes

Reviewer #2: Partly

3. Has the statistical analysis been performed appropriately and rigorously? 

Reviewer #1: Yes

Reviewer #2: (No Response)

4. Have the authors made all data underlying the findings in their manuscript fully available?

Reviewer #1: Yes

Reviewer #2: No

5. Is the manuscript presented in an intelligible fashion and written in standard English?

Reviewer #1: Yes

Reviewer #2: Yes

6. Review Comments to the Author

Reviewer #1: There are minor corrections such as scientific names of organisms not italicized in accordance with general taxonomy rules, kindly check all the scientific names in the manuscript and ensure that they are in italics.

Reviewer #2: Study overview

The authors are commended for thoroughly addressing the comments from the initial round of review. The revision has notably improved the overall quality and presentation of the manuscript, congratulations on this progress.

The central idea of the study remains valuable and of high relevance, however, ONE MAJOR concern persists regarding the lack of key information necessary to adequately test the promised objective to “evaluate the prevalence and associated risk factors of H. pylori and S. mansoni co-infection …”.

The study could be further strengthened by including specific additional participant-level information, such as hygiene-related and socio-environmental factors etc, probably from the bigger project. Incorporating these variables would allow for a more robust analysis, richer results and discussion, and ultimately a much stronger manuscript with greater potential to inform policy decisions.

Major comments

1.The statistical analysis and the lack of key data necessary to strengthen the manuscript’s objective are somewhat concerning. For example, the analysis appears to rely primarily on limited demographic variables (age and sex only), which are insufficient to fully address the stated objective. Crucial participant-level information that would allow for a more robust evaluation of potential risk factors, as outlined in the objective, is missing.

Similarly, the exclusive use of univariate logistic regression limits the analytical depth of the study. While this approach can identify relationships/associations between individual independent variables and the outcome variable, it does not account for confounding factors that may influence these relationships. As such, conducting a multivariable regression analysis that adjusts for relevant confounders is strongly recommended, as this would greatly enhance the analytical rigor and overall quality of the manuscript.

If additional data cannot be obtained or a multivariable analysis cannot be performed, it is suggested that the authors revise that study objective to clarify that the analysis is purely descriptive and limited to evaluating associations between the two pathogens and basic demographic factors. However, this alternative would somewhat reduce the manuscript’s scientific strength.

2.Consider splitting and refining/improving the second paragraph to (i) dedicate one paragraph specifically to H. pylori similar to how you focused on Schistosoma spp. in the first paragraph, (ii) use the subsequent paragraph to discuss the potential co-infection between both pathogens, which would help enhance the contextual depth and clarity of the manuscript.

3.Consider rephrasing “... with about 80% of African population affected [13]” to clarify that a portion of the 80% represents asymptomatic individuals, thus reflecting previous exposure (sero-prevalence) to enhance clarity.

4.The manuscript contains some important limitations which are not discussed in the manuscript. Kindly consider including a limitation section to comprehensively discuss this.

Minor comments

1.It appears that some of the previous comments, specifically minor comments 1, 22 & 24, were not implemented in the revised manuscript. Kindly consider giving due consideration to these important points.

2.Please verify that the correct supplementary table has been uploaded, as the current version appears to contain missing participant data from line 301 onward. Additionally, the lack of standardized variable names and the absence of a codebook describing these variables make the dataset somewhat difficult to interpret. It is also worth reiterating that the R scripts used for the analysis have not been deposited in an open-access repository, which would aid a more comprehensive review and promote reproducible science for researchers interested in conducting similar studies in the future.

3. Although the acronym “WASH” was introduced in the abstract, it would be appropriate to reintroduce it at the first mention in the main text before subsequent use.

4.Please italicize Schistosoma at line 54, and ensure that all generic and specific names are consistently italicized throughout the manuscript for scientific accuracy.

5.The use of “GC” in line 69 is unnecessary, as the term seems to appear only once throughout the manuscript.

6.The use of “N” and “%” after each variable is unnecessary in the Tables, since a column is already dedicated to “N”. Instead, I guess you may include a footnote stating, for example, “values correspond to N (%)”. The same approach could be applied for Tables 2 and 3)

7. PLOS authors have the option to publish the peer review history of their article (what does this mean?). If published, this will include your full peer review and any attached files.

Reviewer #1: **Yes: **Obiageli Josephine Nebe

Reviewer #2: **Yes: **Adedayo Michael Awoniyi

---

## [Author Response · Author response to Decision Letter 2]

20 Oct 2025

RESPONSE TO REVIEWERS _ROUND 2

Reviewer #1:

There are minor corrections such as scientific names of organisms not italicized in accordance with general taxonomy rules, kindly check all the scientific names in the manuscript and ensure that they are in italics.

Response: We thank the reviewer for this comment, and have carefully revised the manuscript to ensure the scientific names are italicized

Reviewer #2:

 Study overview

The authors are commended for thoroughly addressing the comments from the initial round of review. The revision has notably improved the overall quality and presentation of the manuscript, congratulations on this progress.

The central idea of the study remains valuable and of high relevance, however, ONE MAJOR concern persists regarding the lack of key information necessary to adequately test the promised objective to “evaluate the prevalence and associated risk factors of H. pylori and S. mansoni co-infection …”. The study could be further strengthened by including specific additional participant-level information, such as hygiene-related and socio-environmental factors etc, probably from the bigger project. Incorporating these variables would allow for a more robust analysis, richer results and discussion, and ultimately a much stronger manuscript with greater potential to inform policy decisions.

Response: We thank the reviewer for this insightful comment and acknowledge the importance of their suggestion. We agree that incorporating additional covariates could enhance the robustness of the models. However, the current study was primarily designed to describe the co-infection patterns of both pathogens and to assess differences based on age and sex characteristics. We have revised Line 92 to clarify this focus and have also added a section under the study’s limitations to acknowledge this constraint.

Major comments

1.The statistical analysis and the lack of key data necessary to strengthen the manuscript’s objective are somewhat concerning. For example, the analysis appears to rely primarily on limited demographic variables (age and sex only), which are insufficient to fully address the stated objective. Crucial participant-level information that would allow for a more robust evaluation of potential risk factors, as outlined in the objective, is missing.

Similarly, the exclusive use of univariate logistic regression limits the analytical depth of the study. While this approach can identify relationships/associations between individual independent variables and the outcome variable, it does not account for confounding factors that may influence these relationships. As such, conducting a multivariable regression analysis that adjusts for relevant confounders is strongly recommended, as this would greatly enhance the analytical rigor and overall quality of the manuscript.

If additional data cannot be obtained or a multivariable analysis cannot be performed, it is suggested that the authors revise that study objective to clarify that the analysis is purely descriptive and limited to evaluating associations between the two pathogens and basic demographic factors. However, this alternative would somewhat reduce the manuscript’s scientific strength.

Response: We appreciate the reviewer’s comment. As stated in the manuscript, our primary objective was to explore associations based on age and gender rather than to develop a comprehensive predictive model, which would require additional covariates. This approach may be considered in future studies. Additionally, since only two covariates were included and one of them (gender) was not significantly associated (p-value > 0.2), including both in a multivariate model would not meaningfully change the interpretation. Furthermore, with no evidence of potential confounding, adjusting for gender would not alter the effect of age. Therefore, the univariate model presented provides a parsimonious and stable estimate of the association.

2.Consider splitting and refining/improving the second paragraph to (i) dedicate one paragraph specifically to H. pylori similar to how you focused on Schistosoma spp. in the first paragraph, (ii) use the subsequent paragraph to discuss the potential co-infection between both pathogens, which would help enhance the contextual depth and clarity of the manuscript.

Response: We thank the reviewer for this comment. Our introduction is currently two pages long, and we aim to keep it within this limit while maintaining a logical flow. The intent was to build a case for improved management of H. pylori infections as part of routine schistosomiasis control programs. To achieve this, we began with a paragraph on schistosomiasis and linked it to H. pylori. The second paragraph focused primarily on H. pylori, while connecting its epidemiology with schistosomiasis. Between lines 77–83, we discussed predilection sites, comorbidities, and speculations on altered disease pathology for both infections, citing six references within this section. We would appreciate it if the reviewer allowed our text to remain as presented.

3.Consider rephrasing “... with about 80% of African population affected [13]” to clarify that a portion of the 80% represents asymptomatic individuals, thus reflecting previous exposure (sero-prevalence) to enhance clarity.

Response: We thank the reviewer for this comment. We have revised the text to indicate that H. pylori infections can be acquired during childhood and that prevalence may reach up to 80%. This revision reflects the fact that H. pylori typically persists chronically after acquisition, usually in early life, and that individuals with detectable antibodies are considered presumptively infected or may require clinical evaluation

4.The manuscript contains some important limitations which are not discussed in the manuscript. Kindly consider including a limitation section to comprehensively discuss this.

Reviewer: we thank the reviewer for this comment, we have now included a limitation section

Minor comments

1.It appears that some of the previous comments, specifically minor comments 1, 22 & 24, were not implemented in the revised manuscript. Kindly consider giving due consideration to these important points.

Response: We thank the reviewer for this comment. Previous comment 22 referred to the use of prevalence ratios (PR) versus odds ratios (OR). We apologize for missing this in the earlier review. Our primary objective was to describe co-infection patterns, which we addressed using basic descriptive statistics. The secondary objective was to examine whether infection patterns were associated with age or gender. In this context, ORs obtained via logistic regression provide a valid measure of association. In contrast, prevalence ratios or risk ratios compare the proportion of individuals with an outcome between groups, providing a measure of relative prevalence rather than the strength of association.

Regarding comment 24, we have provided the dataset as a supplementary file and included the analysis code. While we are not obliged to provide the scripts, readers interested in obtaining them can make a reasonable request to the corresponding author. Lastly, we would like to retain the list of keywords used

2.Please verify that the correct supplementary table has been uploaded, as the current version appears to contain missing participant data from line 301 onward. Additionally, the lack of standardized variable names and the absence of a codebook describing these variables make the dataset somewhat difficult to interpret. It is also worth reiterating that the R scripts used for the analysis have not been deposited in an open-access repository, which would aid a more comprehensive review and promote reproducible science for researchers interested in conducting similar studies in the future.

Response: we have provided the dataset as a supplementary file and included the analysis code. While we are not obliged to provide the scripts, readers interested in obtaining them can make a reasonable request to the corresponding author.

3. Although the acronym “WASH” was introduced in the abstract, it would be appropriate to reintroduce it at the first mention in the main text before subsequent use.

Response: we thank the reviewer for this comment. We have revised it, please see line 105

4.Please italicize Schistosoma at line 54, and ensure that all generic and specific names are consistently italicized throughout the manuscript for scientific accuracy.

Response: we thank the reviewer for this comment. We have revised it,

5.The use of “GC” in line 69 is unnecessary, as the term seems to appear only once throughout the manuscript.

Response: we thank the reviewer for this comment. We have revised it,

6.The use of “N” and “%” after each variable is unnecessary in the Tables, since a column is already dedicated to “N”. Instead, I guess you may include a footnote stating, for example, “values correspond to N (%)”. The same approach could be applied for Tables 2 and 3)

Response: we thank the reviewer for this comment. We have revised it,

---

## [Decision Letter · Decision Letter 2]

10 Nov 2025

Co-infections of Schistosoma mansoni and Helicobacter pylori in School-Aged Populations and implication for management and control practices in Niger State, Nigeria

PONE-D-25-40863R2

Dear Dr. Mogaji,

We’re pleased to inform you that your manuscript has been judged scientifically suitable for publication and will be formally accepted for publication once it meets all outstanding technical requirements.

Kind regards,

Clement Ameh Yaro, Ph.D

Academic Editor

PLOS ONE

Additional Editor Comments (optional):

Reviewers' comments:

Reviewer's Responses to Questions

**Comments to the Author**

1. If the authors have adequately addressed your comments raised in a previous round of review and you feel that this manuscript is now acceptable for publication, you may indicate that here to bypass the “Comments to the Author” section, enter your conflict of interest statement in the “Confidential to Editor” section, and submit your "Accept" recommendation.

Reviewer #2: (No Response)

2. Is the manuscript technically sound, and do the data support the conclusions?

Reviewer #2: Partly

3. Has the statistical analysis been performed appropriately and rigorously? 

Reviewer #2: (No Response)

4. Have the authors made all data underlying the findings in their manuscript fully available?

Reviewer #2: No

5. Is the manuscript presented in an intelligible fashion and written in standard English?

Reviewer #2: Yes

6. Review Comments to the Author

Reviewer #2: (No Response)

7. PLOS authors have the option to publish the peer review history of their article (what does this mean?). If published, this will include your full peer review and any attached files.

Reviewer #2: No

---

## [Editor Report · Acceptance letter]

PONE-D-25-40863R2

PLOS ONE

Dear Dr. Mogaji,

I'm pleased to inform you that your manuscript has been deemed suitable for publication in PLOS ONE. Congratulations! Your manuscript is now being handed over to our production team.

Kind regards,

on behalf of

Dr. Clement Ameh Yaro

Academic Editor

PLOS ONE